# Differential Gene and Protein Expression of Conjunctival Bleb Hyperfibrosis in Early Failure of Glaucoma Surgery

**DOI:** 10.3390/ijms241511949

**Published:** 2023-07-26

**Authors:** Elena Millá, Néstor Ventura-Abreu, Cristina Vendrell, Maria Jesús Muniesa, Marta Pazos, Xavier Gasull, Núria Comes

**Affiliations:** 1Hospital Clínic de Barcelona, Institut Clinic d’Oftalmologia, ICOF, Sabino Arana nº1, 08028 Barcelona, Spain; emilla@clinic.cat (E.M.); mariajesus.muniesa@gmail.com (M.J.M.); pazos@clinic.cat (M.P.); 2Institut Comtal d’Oftalmologia, Innova Ocular-ICO Barcelona, Via Augusta 48, 08006 Barcelona, Spain; cvendrell@icoftalmologia.es; 3Hospital Universitari Sagrat Cor., Viladomat 288, 08029 Barcelona, Spain; ventanes@gmail.com; 4Hospital de Viladecans, Avda. Gavà 38, 08840 Barcelona, Spain; 5Neurophysiology Laboratory, Department of Biomedicine, Medical School, University of Barcelona, Casanova 143, 08036 Barcelona, Spain; xgasull@ub.edu; 6Institute of Neurosciences, University of Barcelona, Edifici de Ponent, 2n vagó 3r pis, Passeig de la Vall d’Hebron 171, 08035 Barcelona, Spain; 7Institut d’Investigacions Biomèdiques August Pi i Sunyer (IDIBAPS), Rosselló 149, 08036 Barcelona, Spain

**Keywords:** conjunctival filtration bleb, glaucoma surgery failure, conjunctival fibrosis, gene expression, profibrotic genes

## Abstract

The early failure of glaucoma surgery is mainly caused by over-fibrosis at the subconjunctival space, causing obliteration of the filtration bleb. Because fibrosis has a suspected basis of genetic predisposition, we have undertaken a prospective study to identify upregulated profibrotic genes in a population of glaucoma patients with signs of conjunctival fibrosis and early postoperative surgical failure. Clinical data of re-operated fibrosis patients, hyperfibrosis patients who re-operated more than once in a short time, and control patients with no fibrosis were recorded and analyzed at each follow-up visit. Conjunctival-Tenon surgical specimens were obtained intraoperatively to evaluate the local expression of a panel of genes potentially associated with fibrosis. In order to correlate gene expression signatures with protein levels, we quantified secreted proteins in primary cultures of fibroblasts from patients. Expression of *VEGFA*, *CXCL8*, *MYC*, and *CDKN1A* was induced in the conjunctiva of hyperfibrosis patients. VEGFA and IL8 protein levels were also increased in fibroblast supernatants. We propose that an increase in these proteins could be useful in detecting conjunctival fibrosis in glaucoma patients undergoing filtering surgery. Molecular markers could be crucial for early detection of patients at high risk of failure of filtration surgery, leading to more optimal and personalized treatments.

## 1. Introduction

Glaucoma is a group of optic neuropathies causing irreversible damage to the optic nerve that leads to progressive and permanent loss of vision and, ultimately, blindness. Elevated intraocular pressure (IOP) is the main risk factor for glaucoma; it is mostly caused by an impairment of the eye’s aqueous humor (AH) through the drainage structures. Primary open-angle glaucoma (POAG) is the most common type of glaucoma in Western countries, and approximately half of glaucoma cases still remain undiagnosed. Early detection and treatment are crucial as they can slow or even stop the progression of the disease. Pharmacological treatment, such as eye drops, is the most common way to treat glaucoma to reduce IOP by decreasing AH production or increasing its outflow. If IOP control is not achieved within a physiological range by pharmacological treatment or it causes excessive undesirable effects, the surgical option is considered. The objective of the surgery is to favor the filtration and drainage of the AH [1]. The most commonly used initial surgical technique is filtering surgery (perforating or not) that seeks to create a fistula from the subconjunctival space towards the anterior chamber of the eye in order to promote the exit of AH through it and a decrease in IOP. Hence, AH is stored in the conjunctival filtration bleb, which is a subconjunctival reservoir, and the success of the surgery depends entirely on the degree of survival and proper functioning of the bleb [2]. An excess of subconjunctival scarring in the postoperative period leads to an early surgery failure when the bleb is obliterated. To minimize post-surgical fibrosis, the usual intraoperative administration of antimetabolites (mitomycin C or 5 fluorouracil) acts by inhibiting the proliferation of fibroblasts present mainly in Tenon’s layer. However, these drugs can, in turn, cause complications such as ocular hypotony due to scleral impregnation or corneal toxicity by acting in a non-specific way [3].

In certain patients, despite using antimetabolites, a marked profibrotic tendency is observed with early changes in the conjunctival filtration bleb, which raises suspicions of its early closure. The cause of this profibrotic tendency could have a double genetic and acquired component. Among the acquired causes would be the prolonged use of hypotensive eye drops containing preservatives like benzalkonium chloride (BAK), which cause long-term toxicity of the ocular surface, and the use of topical prostaglandin analogs that cause chronic hyperemia. This chronic inflammatory infiltration, located in the conjunctival epithelium and connective tissue, increases the fibroblast population as well as collagen production, causing bleb obliteration. This toxic effect has been confirmed through impression cytology, immunohistochemistry, and in vivo confocal microscopy, and the intensity of these changes is related to the amount of preservative administered and the duration of treatment [4]. Likewise, patients who have undergone repeated eye surgeries or with previous inflammation conditions such as uveitis show a greater tendency to develop fibrosis and scarring of the conjunctival filtration bleb. In patients with a high degree of preoperative inflammation, pre and postoperative measures should be intensified to prevent the fibrosis that leads to surgical failure. These actions would consist of withdrawing hypotensive drugs with preservatives, using preoperative topical anti-inflammatories to reduce the degree of conjunctival inflammation, or increasing antimitotic treatment, among others [5]. However, in some patients, in spite of following those preoperative measures to treat the ocular surface before surgery, a tendency towards early bleb fibrosis is observed. As a result, surgical failure occurs even if some postoperative maneuvers are performed, like needling or subconjunctival injections of antimetabolites. Therefore, apart from the acquired profibrotic factors, it is suspected that the individual susceptibility to early fibrosis of the filtration bleb could be driven by changes in the conjunctival gene expression patterns. Interestingly, silencing of specific profibrotic genes has been effective in modulating fibroblast functions [6] and reducing the healing rate [7]; as a result, it may increase the survival of the conjunctival filtration bleb to improve long-term glaucoma surgery [8].

In essence, scarring and fibrosis can be considered an overreach of the healing process. Wound healing is a dynamic physiological mechanism consisting of different overlapping phases activated after tissue damage from disease, injury, or surgery. The incision of glaucoma filtering surgery initially causes damage to the conjunctiva and to surrounding blood vessels, which initiates the inflammatory phase. At this stage, the exposure of subendothelial collagen, the presence of extracellular matrix (ECM) material, and the extravasation of plasma proteins activate homeostasis. It includes the formation of a platelet plug, activation of the coagulation cascade, and generation of the blood clot to maintain the integrity of the vasculature [5,9]. The inflammatory phase is characterized by the invasion of neutrophils, monocytes, and macrophages that release different growth factors, including EGF, VEGFA, TGFB1, PDGF, CSF2, and FGF and inflammatory cytokines such as IL1A, IL6, IL8, and TNFA [10,11]. Some mediators are involved in the maintenance and subsequent activation of fibroblasts that play an essential role in the production of granulation tissue and angiogenesis during the proliferative phase [12,13]. Upon differentiation to contractile myofibroblasts, they also promote the production of collagen and other components of the ECM and, as a result, the contraction and the closure of the wound. The final stage of the remodeling phase consists of the transition between granulation tissue and scar tissue through selective degradation of the ECM and myofibroblast apoptosis [14,15]. Dysregulation of this complex network of molecular pathways could contribute to the persistent presence of myofibroblasts and to abnormal ECM deposition that characterizes the fibrotic response associated with numerous diseases [16].

In this study, we have prospectively evaluated the local expression of a panel of genes potentially associated with the development of fibrosis and the secreted proteins they encode. We analyzed a cohort of glaucoma patients who required more than one surgical intervention in a short period of time due to early surgery failure caused by conjunctival scarring in order to define a hyperfibrosis genotype common to all of them. For this, we have established the genotype-phenotype correlation between quantified gene/protein expression and the clinical evolution of the patients toward the success or failure of surgery. Knowing which profibrotic genes have altered expression in patients with early surgical failure may allow us to identify patients with a higher risk of conjunctival fibrosis. Our ultimate goal is to detect cases at high risk of scarring that could compromise surgery and accordingly personalize the type of surgery offered.

## 2. Results

### 2.1. Patient Demographics

Eighty-eight patients were prospectively included in the study in which we carried out a two-year clinical follow-up of a cohort of glaucoma patients who underwent surgery, taking tissue samples from control patients at the first surgery and from patients with fibrosis who had to be re-operated (fibrotic/hyperfibrotic patients) at the second surgery (first revision). We compared conjunctival gene expression of fibrosis and hyperfibrosis patients with the control group who never developed ocular fibrosis (successfully operated only once) in order to identify the genetic factors implicated in the conjunctival wound healing process and the early failure of the filtering bleb. The patients’ demography is displayed in Table 1. There were no statistically significant differences regarding age, eye laterality, sex, preoperative IOP level, preoperative glaucoma medications, and glaucoma severity (degree of mean visual field defect) between the fibrosis and no fibrosis groups. Only the proportion of PACG was significantly higher in the non-fibrosis group, compared to the “others” subgroup (namely post-traumatic, neovascular, or post-surgical glaucoma, which was more frequent in the fibrosis group). The fibrosis group had a higher proportion of individuals with pseudoexfoliation glaucoma, which is known for being a more severe type of glaucoma with higher preoperative IOP, a more complex surgery if phacoemulsification was also performed and a higher degree of postoperative intraocular inflammation. There was a statistically significant higher proportion of pseudophakic patients among the fibrosis group. Table 2 shows the data related to the surgical procedures performed as well as the intraoperative or postoperative maneuvers required. Regarding the type of surgery, the fibrosis group had a higher proportion of patients who operated with a drainage device implantation (Ahmed^®^, Baerveldt^®,^ or Paul^®^ implants), but none of the comparisons were statistically significant between non-fibrosis and fibrosis patients. In addition, when analyzing the presence of previous glaucoma surgeries, they were clearly more frequent in the fibrosis group, which also needed more postoperative maneuvers such as needlings and major revisions of the bleb. Finally, the fibrosis group required a higher (yet not significant) more frequent use of wound healing modulators such as antimetabolites or the Ologen^®^ implant.

### 2.2. Gene Expression Analysis

We examined expression levels of a panel of genes potentially involved in the development of fibrosis in glaucoma patients who had undergone filtering surgery. Thus, we quantified the mRNA expression of *VEGFA*, *TGFA*, *TGFB1*, *CXCL8*, *IL18*, *MYC*, *THBS1*, *CDKN1A*, and *CDKN2A* by quantitative real-time PCR (the complete name of the genes are in Abbreviations). All genes were found expressed at the conjunctival level except *CCN2*, which showed insufficient expression levels to be quantified. When the gene expression was determined in fibrosis patients (n = 41) compared to the control group (n = 32), a slight trend towards upregulation in *VEGFA* (fold change = 1.56, *p* = 0.0617), *CXCL8* (fold change = 1.65, *p* = 0.1009), and *CDKN1A* (fold change = 2.13, *p* = 0.0148) was observed, without being statistically significant with the exception of the *CDKN1A* gene. Instead, the expression of *TGFB1*, *IL18*, *MYC*, *THBS1*, and *CDKN2A* was not altered or was very slightly altered (fold change ≤ 1.5) in fibrosis versus control patients. Next, we examined local gene expression in a group of patients who were re-operated more than once due to marked conjunctival fibrosis signs before the first postoperative month (hyperfibrosis patients, n = 6). When compared to controls, we found a significant overexpression of *VEGFA* (fold change = 6.47, *p* = 0.0001), *CXCL8* (fold change = 7.32, *p* = 0.0003), *MYC* (fold change = 5.79, *p* = 0.0006), and *CDKN1A* (fold change = 6.20, *p* = 0.0059) genes (Table 3, Figure 1a). Hence, *VEGFA*, *CXCL8*, and *CDKN1A*, with a mild tendency to be upregulated in fibrosis patients, were markedly induced in the hyperfibrosis ones, with more pronounced signs of conjunctival fibrosis and very early failure of filtering surgery.

### 2.3. Protein Levels of Secreted Proteins

VEGFA, IL8, TFGB1, and TSP1 protein levels were measured in the culture medium of conjunctival fibroblasts from hyperfibrosis patients and control patients by ELISA (n = 5 control, n = 6 hyperfibrosis). We previously found the *VEGFA* and *CXCL8* (encodes IL8) genes significantly upregulated in the conjunctiva of hyperfibrosis glaucoma patients, unlike *TGFB1* and *THBS1* (encodes TSP1), which were used as potential controls of non-induced proteins. The findings demonstrated that, despite individual differences, levels of the soluble proteins VEGFA and IL8 were significantly increased in hyperfibrosis patients compared with controls (*p* = 0.0007 and *p* ≤ 0.0001, respectively). In contrast, TGFB1 and TSP1 levels were not altered in fibroblast supernatants from hyperfibrosis patients versus control patients (*p* = 0.2995 and *p* = 0.6894, respectively) (Table 4, Figure 2). Thus, the obtained results suggested that the increased secretion of IL8 and VEGFA proteins could be indicative of conjunctival fibrosis associated with glaucoma filtering surgery.

## 3. Discussion

Conjunctival fibrosis of the filtering bleb is the result of an excessive wound-healing response and is one of the main causes of early failure of glaucoma surgery. Despite the strategies used to deal with the acquired factors that predispose to fibrosis, including treatment with antimetabolites and preoperative ocular surface preparation, a number of patients develop early postoperative conjunctival fibrosis [5]. Thus, the profibrotic tendency has a genetic origin that would explain why some patients whose a priori do not present signs or risk factors leading to exaggerated healing develop an inexorable scarring at the level of the subconjunctival space that ends in surgical failure. In recent years, the presence of genetic factors and variants has been identified in patients who present hyperfibrotic phenomena in different regions of the body [17,18]. These patients show a great tendency to exhibit hypertrophic or keloid-type scarring skin lesions and to suffer pathologies, including glomerulosclerosis and sarcoidosis [19,20].

With this study, we aim to contribute to identifying the genes involved in the development of conjunctival fibrosis. To pursue that end, we quantified the expression levels of nine potentially relevant genes in fibrosis in the conjunctiva (Tenon’s capsule) from glaucoma patients who underwent filtration surgery. Among them, we identified individuals with a clear hyperfibrosis phenotype, in whom surgery failed more than once in a short time, others with a fibrosis phenotype that had to be re-operated one time and with a slow fibrosis evolution, and those that did not present fibrosis and showed surgical success. Clinically, there were some statistically significant differences between patients with bleb fibrosis and the control (no fibrosis) group. Fibrosis patients presented a history of having previous glaucoma surgeries, a need for more postoperative surgical maneuvers, and the use of more wound healing modulator agents than the non-fibrosis patients. They showed more complex types of glaucoma, such as the pseudoexfoliative one, and required more aggressive surgeries like drainage device implantation as well. However, no differences were found regarding age, sex, eye laterality, preoperative IOP, glaucoma staging severity, or number of drugs in the preoperative hypotensive therapeutic regimen. Paradoxically, no fibrosis patients presented a significantly higher proportion of preoperative usage of BAK-preserved drugs. This may be due to the fact that patients who have already had a decreased effect of their filtering surgery and need additional postoperative medication are treated more cautiously with drugs without BAK so as not to contribute to increasing conjunctival fibrosis and, with it, the complete failure of the bleb.

Recent studies revealed changes in conjunctival gene expression associated with fibrosis after glaucoma filtration surgery. In a pivotal study in which expression of 88 genes previously associated with wound healing was quantified in the conjunctiva of glaucoma patients, significant differences were found in 29 genes at an early stage after glaucoma surgery, 20 genes in the next 90–180 days, and 2 genes one year after surgery. During the postoperative period (15 days), *CXCL8*, *VEGFA*, *CDKN2A*, *CDKN1A*, and *TGFA* were the most upregulated genes, while others, like *IL18* and *MYC,* were downregulated. Instead, at 90 days after surgery, a noteworthy number of genes, such as *HIF1A*, *EGFR*, *MYC*, *TGFB1*, *TGFB2*, *BCL2*, *MMP2,* and *PDGF*, among others, were downregulated. One year after surgery, however, only the expression of *TGFB1* and *ITGB2* was reduced. Additionally, in peripheral blood, the most evident changes occurred also during the first 15 days after surgery, when 12 genes (*IL1B*, *CEBPD*, *HIF1A*, *ITGB2*, *TNFRSF1A,* and *CASP1*, among others) were induced [21]. These results suggested that conjunctival fibrosis developed after filtration surgery could have a significant genetic component driven by an altered expression of specific genes. In addition, positive regulation of many of the fibrosis-related genes seems to occur in the first few days after surgery. In our study, *VEGFA* and *CXCL8*, in conjunction with *CDKN1A*, were the most upregulated genes in fibrosis patients. Interestingly, these changes were much more pronounced in patients with hyperfibrosis in whom *VEGFA*, *CXCL8*, *CDKN1A,* and *MYC* were significantly induced. Therefore, this altered expression pattern may be relevant in the progression of fibrosis and the surgical outcome in glaucoma. More patients should be analyzed to corroborate whether the degree of the upregulation of these genes could be related to the severity of conjunctival fibrosis.

Likewise, a previous correspondence analysis revealed *VEGFA* as the only tested gene differentially expressed in patients with surgical failure versus patients with successful surgery [21]. VEGFA is a glycoprotein highly expressed in acute wounds [22] that has been involved in many processes linked to wound healing repair. It increases the rate of proliferation [23] and migration [24] of endothelial cells and promotes wound angiogenesis [25] and the formation of lymphatic vessels [26]. Several studies have reported a reduction in vascularity and fibrosis during the wound healing process by neutralization of VEGFA [27]. Hence, anti-VEGFA monoclonal antibodies (bevacizumab) reduce the proliferation of cultured Tenon fibroblast, collagen deposition, and angiogenesis in a rabbit model of trabeculectomy, improving surgical outcome [28]. Pilot studies have also shown that the administration of bevacizumab decreases IOP in patients with neovascular glaucoma [29]. Similarly to *VEGFA*, the chemokine *CXCL8* is upregulated in the acute phase of wound healing [30]. It plays a key role in neutrophils and T-lymphocytes chemotaxis in the inflammatory response [31]. IL8 also increases proliferation and migration of endothelial cells as well as wound angiogenesis [32,33]. Our results, in the same line as those of others [21], show that overexpression of *VEGFA* and *CXCL8* could be used as an indicator of conjunctival fibrosis following filtering surgery.

Our data also revealed an obvious dysregulation of *MYC* and *CDKN1A* expression in the conjunctiva of glaucoma patients with hyperfibrosis. Both genes involved in the cell cycle control were found to be significantly overexpressed. *CDKN1A* was previously reported to be upregulated in the conjunctiva during the postoperative period of filtering surgery, although the expression of *MYC* was reduced [21]. It is known that the *MYC* oncogene promotes cell growth by counteracting the action of several negative regulators of the cell cycle [34,35], such as *CDKN1A,* in a *TGFB1*-dependent manner [36]. Consistently, the downregulation of *MYC* with specific antisense RNA decreases the progression of the cell cycle [37]. Interestingly, several studies suggested that the MYC proto-oncogene protein could be used as a molecular marker of impaired healing in chronic wounds [38]. On the other hand, *CDKN1A* encodes for the CDN1A protein, a cyclin-dependent kinase inhibitor, that has been associated with cell cycle arrest. Induction of CDN1A inhibits cell proliferation and reduces the progression of some tumors [39]. Thus, the upregulation of *CDKN1A* in the conjunctiva could be part of a response mechanism to counteract fibrosis. In addition to elevated expressions of *CDKN1A* in fibrotic tissues [40], genetic variants of *CDKN1A* have also been identified as associated with a greater risk of suffering from idiopathic pulmonary fibrosis [41].

As with other profibrotic genes, *TGFB1* is abundantly expressed during wound healing, where it seems to be involved in all stages. It stimulates fibroblast activation, the synthesis and deposition of collagen I and fibronectin, in addition to the formation of granulation tissue [42]. At lower concentrations, it has been described that TGFB1 induces the proliferation and migration of fibroblasts from Tenon’s capsule [43] and its transformation to myofibroblasts [44,45]. Its potent ability to enhance inflammation, angiogenesis, and collagen production has placed it as one of the key factors in the pathophysiology of ocular fibrosis [46]. Although we did not find changes in *TGFB1* expression in the conjunctiva of patients with hyperfibrosis, it could perform its function in response to wound healing. It is important to point out that when *TGFB1* is induced, it stimulates the expression of *CCN2,* which plays an essential function in the scarring response. *CCN2* is highly expressed in some fibrotic tissues [11], enhancing the proliferation and differentiation of Tenon’s capsule fibroblasts and promoting the deposition of extracellular matrix (ECM) components [47]. In our hands, we could not amplify *CTFG* in conjunctiva tissue samples from patients without fibrosis, probably due to its low expression. We focused our investigation in a very specific panel of genes representative of the different phases of the fibrotic response. Thus, we quantified genes coding for inflammatory cytokines (*CXCL8*, *IL18*), growth factors (*VEGFA*, *TGFA*, *TGFB1*), genes involved in cell cycle control (*MYC*, *CDKN1A*, *CDKN2A*) [21,48] and tissue repair (*THBS1*) [49,50]. In another study, in which a panel of one hundred genes was examined, a distinct group of differentially expressed genes was identified in fibrotic conjunctival fibroblasts from glaucoma patients with previous surgery versus non-fibrotic fibroblasts from patients without previous surgery. The authors found *MYOCD*, *LMO3*, *IL6*, and *RELB* significantly upregulated and *PRG4*, *CD34*, *IL33*, *MMP10*, *CCN5*, *COL6A6*, and *IGFBP5* markedly downregulated in fibrotic fibroblasts. This definitely broadened the list of genes potentially associated with the development of post-surgery conjunctival fibrosis and supports inflammation, ECM remodeling, smooth muscle contraction, and oncogene expression as biological pathways underlying the fibrotic process [48].

In order to correlate the altered gene expression in fibrosis with protein levels and to verify that secreted protein levels were experimentally measurable, we quantified VEGFA, IL8, TGFB1, and THBS1 in the supernatants of cultured fibroblasts from patients. As in gene expression, and despite individual differences, we found a significant increase in secreted VEGFA and IL8 in fibroblasts of patients with hyperfibrosis versus control patients. In contrast, we did not find significant changes in TGFB1 and THBS1 as occurred with the genes coding these proteins. Although there is strong evidence for using mRNA quantification to determine the relative importance of genes in a given process [51], it is convenient to corroborate gene expression changes by measuring protein levels. Given that after a filtering surgery, the AH is drained to the subconjunctival tissue, quantification of specific proteins associated with fibrosis in the AH could be crucial to identify patients at risk of surgical failure. Indeed, high levels of IL8, VEGFA, PDGF, TGFB1, and TGFA have been found in AH of glaucoma patients [52,53]. Because VEGFA neutralization with bevacizumab reduced excessive scarring after glaucoma surgery [28], elevated protein levels of VEGFA in the AH could be a useful indicator of poor surgical outcomes in patients. At present, it is possible to quantify protein levels in tears in non-invasive procedures, and the tear protein profile is being studied for the diagnosis of uveitis [54] and dry eye [55]. The identification of a valid profibrotic marker in tears or by impression cytology in the preoperative period could help us to personalize the surgical treatment of our patients, to choose a non-filtering procedure in those with a high risk of early fibrosis or to adjust the dose and the type of antimetabolites used.

Our objective was to shed light on the identification of crucial genes in the development of conjunctival fibrosis that could be useful for its diagnosis and treatment. We believe that one important contribution of our study was the careful ocular surface preparation in the preoperative period in order to eliminate the bias of the preoperative inflammation that is closely linked with postoperative fibrosis and can, per se, trigger the expression of profibrotic conjunctival gene expression by itself. Under the same conditions, we are recruiting glaucoma patients with different results, from filtering surgery to collecting tear samples. In the near future, we intend to quantify fibrosis markers with techniques that allow us to test a large number of patients. Finally, it is noteworthy that gene silencing of distinct profibrotic agents has been found effective in decreasing the healing rate of the postoperative filtration bleb [56]. A further understanding of the cellular and molecular basis of ocular scarring would contribute to the design of new pharmacological treatments and gene-based therapies for conjunctival fibrosis.

## 4. Materials and Methods

### 4.1. Patients

We included 90 conjunctival tissue samples from 88 individuals (n = 37 control, n = 41 fibrosis, and n = 12 hyperfibrosis), of which 42 (47.72%) were female and 46 were male, with a mean age of 71.10 ± 9.45 years. Patients were recruited from the Institut Clínic d’Oftalmologia (ICOF) at Hospital Clínic de Barcelona, Innova Ocular-ICO, and Hospital de Viladecans (Barcelona, Spain). All analyzed patients were scheduled for glaucoma surgery alone or in combination with cataract surgery. As it is our routine practice, all patients underwent preoperative ocular surface preparation with one or more of the following steps according to the degree of ocular surface disease present. Patients were treated with topical anti-inflammatory treatment (mild steroids, cyclosporine, ectoine), artificial tears with hyaluronate acid, palpebral hygiene, and the use of topical hypotensive drugs with preservatives were substituted by its preservative-free counterparts at least two weeks before surgery. Further, prostaglandin analogs were withdrawn one or two weeks before surgery if possible, and in some cases, oral carbonic anhydrase inhibitors were administered.

The exclusion criteria were patients affected by any kind of active intraocular inflammation such as chronic uveitis, history of severe ocular comorbidities, or several previous interventions (retinal, corneal pathologies, and surgeries, among others) that could greatly increase the likelihood of glaucoma surgical failure. Clinically, patients were classified into three groups after surgery according to the degree of surgical failure caused by the presence of early conjunctival fibrosis. It was diagnosed by slit lamp evaluation of the patient, optical coherence tomography (OCT) testing of the conjunctival bleb, and finding an inadequate level of IOP that required secondary surgical procedures due to bleb fibrosis.

### 4.2. Experimental Groups

Glaucoma patients were classified into three experimental groups: control, fibrosis, and hyperfibrosis patients. Control patients had successfully undergone a single glaucoma surgery, as they did not develop conjunctival fibrosis, at least during the first year of follow-up. Patients were considered non-fibrosis patients if they presented a favorable postoperative evolution with no morphologic nor tomographic signs of bleb fibrosis (diffuse, non-vascularized bleb, presence of microcystins, etc.). On the other hand, the fibrosis group presented signs of bleb fibrosis (flattened, corkscrew vessels, Tenon cyst, etc.) from the first postoperative month onwards that led to the failure of glaucoma surgery and had to be re-operated after the first postoperative month. Finally, the hyperfibrosis group of patients included those who showed signs of surgical failure due to early bleb scarring before the first postoperative month and needed one or more surgical maneuvers/re-interventions in a short period of time. We quantified conjunctival gene expression (n = 32 control, n = 41 fibrosis, and n = 6 hyperfibrosis) and protein levels from supernatants of primary cultures of the patient’s fibroblast (n = 5 control, and n = 6 hyperfibrosis). Two glaucoma patients with hyperfibrosis could be analyzed by PCR and ELISA, and they are counted in both groups. For the demographic statistical analysis, the hyperfibrosis patients were included in the fibrosis group (Table 1).

### 4.3. Study Design

We conducted a prospective, interventional study of the wound-healing gene expression and protein expression of 86 different patients who underwent glaucoma surgery (Table 1). The analysis included the quantification of the expression of genes potentially associated with conjunctival fibrosis in tissue samples and of the secreted proteins’ levels in primary cultures of conjunctival fibroblasts. Patients affected with medically uncontrolled glaucoma or with a previous filtering glaucoma procedure were scheduled for glaucoma surgery, and after receiving preoperative ocular surface preparation (described above), they were operated with different glaucoma techniques according to the characteristics of their glaucoma (Table 2). From each patient, a single conjunctival tissue sample was collected during the glaucoma operation, performed by two surgeons (EM, NV-A) who used identical pre- and postoperative protocols and surgical techniques.

Clinical data of the patients were recorded preoperatively, intraoperatively, and at each follow-up visit (24 h, 1 week, 1, 3, 6 months, and 1 year) in order to detect early bleb fibrosis. Data were collected from the complete preoperative ophthalmological examination that included visual acuity using Snellen optotypes; refraction with auto-refractometer; intraocular pressure by Goldmann applanation tonometer; ultrasound pachymetry; gonioscopy with a gonioscopy lens; fundus by direct visualization of the optic disc and retinography with a non-mydriatic camera; visual field with Sita Standard 24-2 protocol with Humphrey perimeter; Optical coherence tomography of the Cirrus optic nerve and ganglion cell layer. The number and duration of previous ocular hypotensive agents, as well as the history of previous ocular pathologies and interventions, were collected. An assessment of the ocular surface status by slit lamp examination and the presence of dry eye disease (corneal fluorescein staining according to Oxford scale, Schirmer test, break up time) were performed in order to determine the type and quantity of preoperative ocular surface preparation needed in each case.

Regarding the surgical data, the type of intervention performed, the prostheses used, the need for the use of antimetabolite drugs, their dose and concentration, and the existence of per-operative complications such as excessive bleeding, tissue fragility, etc., were annotated. At each postoperative control visit, a complete ophthalmological examination was carried out with the tests mentioned above, and, in particular, a photograph of the filtration bleb with the anterior pole camera and an examination of its internal structure by means of optical coherence tomography. Swept-Source (Triton, Topcon Healthcare, Oakland, NJ, USA) was used in order to anticipate early conjunctival failure as early as possible. If signs of early failure of the conjunctival bleb were observed, postoperative maneuvers were carried out to inhibit or decrease fibrosis at this level, such as subconjunctival punctures of antimetabolites (5-fluorouracil or mitomycin C), needlings with a 30 G needle in the slit lamp or in the operating room, surgical revisions of the bleb or new surgeries with a matrix implant of subconjunctival collagen Ologen^®^ (Equipsa, Madrid, Spain).

### 4.4. Sample Collection and Primary Cell Cultures

During each intervention, a block of conjunctival and Tenon tissue with a surface area of 2 mm × 2 mm was obtained from glaucoma patients. Tissues intended to study local gene expression by quantitative real-time PCR were maintained in RNAlater^®^ (Invitrogen, ThermoFisher Scientific, Waltham, MA, USA) at room temperature for two days to stabilize RNA until extraction. For protein quantification in supernatants from cultured fibroblasts, tissue samples were kept in complete low glucose DMEM supplemented with 20% FBS, 100 mg/mL, L-glutamine, 100 U.I./mL penicillin and 100 μg/mL streptomycin for a maximum of 2 h until primary cell cultures were performed. Conjunctival fibroblasts were obtained using the explant method. Briefly, tissue was cut into small pieces, carefully attached to the bottom of a 35 mm dish, and covered with the same DMEM medium under a glass coverslip. Dishes were incubated in a humidified 5% CO_2_ atmosphere at 37 °C, and the medium was changed every other day. Cells were allowed to grow at confluence from the explant for a period of 2 weeks. Fibroblasts were passaged with Trypsin-EDTA to a T-25 flask and were maintained in the same medium with 10% FBS. Cell supernatants were collected on day 7 of this first passage. All reagents were obtained from Sigma (Madrid, Spain).

### 4.5. RNA Extraction and Reverse Transcription Reaction

Total RNA extraction was conducted with the NucleoSpin RNA from Macherey-Nagel (Düren, Germany). Reverse transcription reactions to generate cDNA were carried out with spectrophotometrically measured RNA (300 ng, Nanodrop, Thermo Fisher Scientific, Waltham, MA, USA) using the SuperScript^TM^ IV First-Strand Synthesis System (Invitrogen, Thermo Fisher Scientific) according to the manufacturer’s recommendations.

### 4.6. Panel of Genes

We have focused the study on a specific group of nine influent genes on the fibrotic response. These genes of interest were selected through a review of the literature according to the following criteria: (I) that, together, represent each of the three phases of the wound healing process (inflammation, proliferation, and remodeling), and (II) that have shown altered conjunctival expression in patients after glaucoma filtering surgery, patients with postoperative fibrosis, or patients with surgical failure, in at least one previous study [21,48]. The only gene not meeting criterion II was THBS1, selected for being highly relevant to the remodeling phase as it has been associated with ECM remodeling in glaucoma, chronic ocular inflammation, and fibrosis development [49,50].

### 4.7. Gene Primers

Primer sets for fibrosis-related genes, as well as the ribosomal 18S RNA, were designed using the Primer Designing Tool of the National Center for Biotechnology Information (NCBI) and purchased from Invitrogen (ThermoFisher Scientific) (Table 5). Target specificity was obtained by performing Basic Local Alignment Search Tool (BLAST) comparisons against the entire nucleotide database (http://www.ncbi.nlm.nih.gov (accessed on 20 February 2019)).

### 4.8. Quantitative Real-Time PCR

Normalized mRNA quantification was performed by relative quantitative real-time PCR technology with Fast SYBR™ Green Master Mix using an ABI Prism^®^ 7500 Fast Real-Time PCR System and the 7500 System SDS v1.5.1 software (Applied Biosystems, ThermoFisher Scientific). Melting curves were carried out to determine melting temperatures in order to check the specificity of the amplified products. All experiments were performed in triplicate. Results are shown as ΔCT; therefore, values are inversely proportional to expression levels. Fold changes between fibrosis/hyperfibrosis versus control samples were calculated by the formula 2^−ΔΔCT^, where CT is the cycle at threshold (automatic measurement), ΔCT is CT of the assayed gene minus CT of the endogenous control (18S rRNA), and ΔΔCT is the ΔCT of the normalized assayed gene in the fibrosis/hyperfibrosis samples minus the ΔCT of the same gene in control one (calibrator). RQ values > 1 correspond to increased fold changes. RQ values < 1 correspond to a fraction of the gene expression and were converted to decreased fold changes by the formula −1/2^−ΔΔCT^.

### 4.9. Protein Quantification by ELISA

VEGFA, IL8, TGFB1, and THBS proteins were determined in cell supernatants using Human ELISA assays (catalog numbers KHG0111, KHC0081, BMS249-4, EHTHBS1, respectively, Invitrogen, ThermoFisher Scientific). Collected DMEM from fibroblast cultures were centrifuged at 18,000 g for 5 min at 4 °C and preserved at −80 °C until quantification. The absorbance was measured at a wavelength of 450 nm using a Spark^®^ multimode microplate reader (Tecan, Männedorf, Switzerland). Two independent experiments with two biological replicates each were performed. All ELISA experiments were carried out according to the manufacturer’s protocols by one technologist.

### 4.10. Statistical Analysis

The Shapiro–Wilk test was used to determine the normality of the distribution. The baseline characteristics were described using the mean ± standard deviation (SD), the median (interquartile range) for quantitative data, and the number (percentage) for categorical variables. Different groups were compared using either an independent *t*-test or Mann–Whitney U test for continuous variables and χ^2^ and Fisher’s exact test for categorical variables as appropriate. In multiple comparisons with significant results, post hoc analysis with Bonferroni’s correction was applied after pairwise comparison. Gene and protein quantification data were presented as the mean ± standard error of the mean (SEM). For all tests, statistical significance was set at a two-tailed *p* value and 95% confidence interval (* *p* < 0.05, ** *p* < 0.01, *** *p* < 0.001, and **** *p* < 0.0001). Statistical tests were performed with GraphPad Prism 9.0 software (GraphPad Software, San Diego, CA, USA) and Stata v14.1 (StataCorp, College Station, TX, USA).

## Figures and Tables

**Figure 1 ijms-24-11949-f001:**
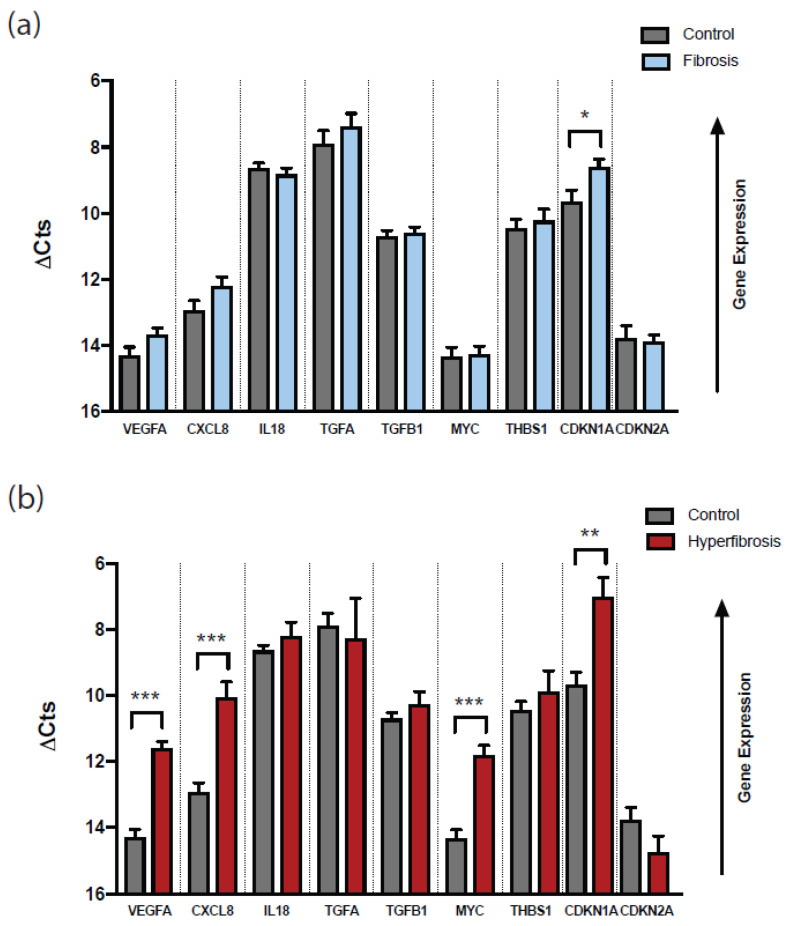
Expression changes of conjunctival genes in fibrosis patients in comparison with non-fibrosis patients. Expression levels of cDNA were measured by relative-quantitative real-time PCR in the conjunctiva of glaucoma patients with signs of fibrosis (fibrosis, hyperfibrosis) and with no fibrosis (control). The expression of all the genes of interest (*VEGFA*, *CXCL8*, *IL18*, *TFGA*, *TFGB1*, *MYC*, *THBS1*, *CDKN1A*, *CDKN2A*) were normalized to *18S* levels to determine changes between conjunctival tissue from (**a**) fibrosis patients that had to be re-operated once (n = 41) and (**b**) hyperfibrosis patients re-operated more than once in a short time (n = 6) versus control patients (n = 32). Data are presented as mean ± SEM of ΔCts of all experiments performed by triplicate (* *p* < 0.05, ** *p* < 0.01, and *** *p* < 0.001 as indicated, Student’s test). Note that the decrease in ΔCt values represents an increase in gene expression.

**Figure 2 ijms-24-11949-f002:**
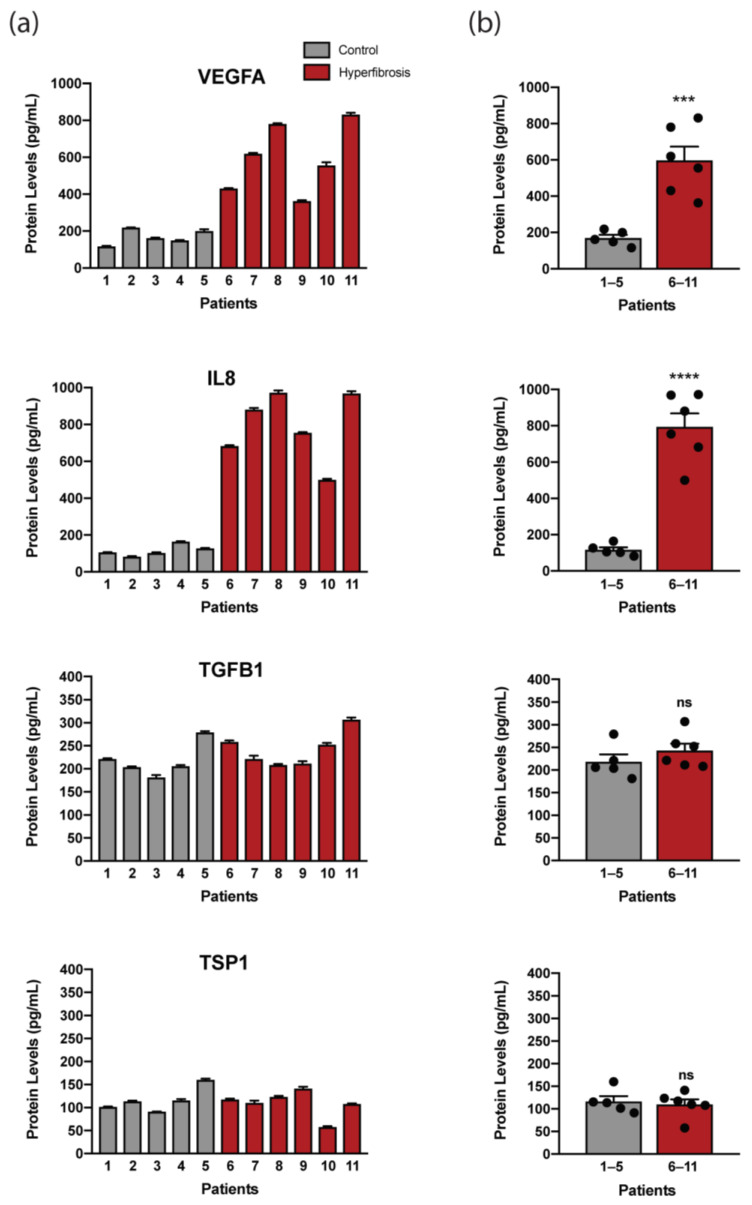
Levels of protein secreted by cultured fibroblasts from fibrosis patients compared with those from non-fibrosis patients. Protein expression levels were quantified in supernatants of primary cultures of conjunctival fibroblasts from patients with no fibrosis (control) and patients with early signs of fibrosis (hyperfibrosis). ELISA was used to measure the levels of VEGFA, IL8, TGFB1, and TSP1 in cell supernatants. (**a**) Protein levels are plotted by a vertical bar for each patient analyzed to show individual differences (1–5 control, 6–11 hyperfibrosis). (**b**) Results are presented as the mean ± SEM in hyperfibrosis patients and non-fibrosis patients of two independent experiments performed by duplicate (ns: non-significant, *** *p* < 0.001, **** *p* < 0.0001 as indicated, Student’s test).

**Table 1 ijms-24-11949-t001:** Patients’ demographics and clinical data.

	No Fibrosis	Fibrosis	*p*-Value
Age (years), mean ± SD	71.3 ± 7.6	70.9 ± 11.3	0.825 *
Gender, n (%) females	18 (50.0)	24 (46.2)	0.722 ^#^
Eye, n (%) right eyes	17 (47.2)	26 (50.0)	0.798 ^#^
**Glaucoma type n (%)**			
POAG	24 (66.7)	34 (65.4)	**0.004** **^+µ^**
PSXG	2 (5.6)	6 (11.5)	
PACG	7 (19.4)	2 (3.9)	
Uveitic	1 (2.8)	3 (5.8)	
Steroid induced	2 (5.6)	0 (0.0)	
Other	0 (0.0)	7 (13.5)	
IOP (mm Hg), mean ± SD	20.9 ± 5.9	23.3 ± 8.1	0.825 *
Glaucoma medications median (IQR)	2 (1.5-3)	2 (1-3)	0.610 *
BAK-preservative drops n (%) yes	25 (69.4)	27 (51.9)	0.125 ^#^
Glaucoma severity (MD), dB median (IQR)	−11.6 [−19.3–(−7.0)]	−14.6 [−21.3–(−7.2)]	0.695 *
Lens status, n (%) pseudophakic	11 (30.6)	31 (59.6)	**0.007 ^#^**

Demographics of patients considering the presence or absence of fibrosis according to clinical appearance. BAK = benzalkonium chloride; dB = decibel; IQR = interquartile range; IOP = intraocular pressure; MD = mean deviation; PACG = primary angle-closure glaucoma; POAG = primary open-angle glaucoma; PSXG = pseudoexfoliative glaucoma; SD = standard deviation. ^µ^ = only the proportion “Steroid-induced” vs. “Other” (*p* = 0.028) remained statistically significant after pairwise comparison using Bonferroni’s correction; * = Mann–Whitney U-Test; ^#^ = χ^2^; ^+^ = Fisher’s exact test. Numbers in bold show statistically significant values.

**Table 2 ijms-24-11949-t002:** Types of surgeries and wound healing treatments.

	No Fibrosis	Fibrosis	*p*-Value
**Type of surgery, n (%)**			
NPDS	21 (58.3)	17 (32.7)	**0.001 ^+$^**
Trabeculectomy	2 (5.6)	1 (1.9)	
ExPRESS^®^	9 (25.0)	5 (8.3)	
XEN45^®^	0 (0.0)	7 (13.5)	
Ahmed^®^ valve	0 (0.0)	2 (3.9)	
Baerveldt^®^ implant	0 (0.0)	9 (17.3)	
PAUL^®^ implant	3 (8.3)	8 (15.4)	
Preseflo^®^	0 (0.0)	1 (1.9)	
iStent^®^	0 (0.0)	1 (1.9)	
Surgical repair (bleb)	1 (2.8)	1 (1.9)	
Previous glaucoma surgeries n (%) yes	0 (0.0)	41 (78.9)	**<0.005 ^#^**
**Type of current surgery n (%)**			
Original	34 (94.4)	30 (57.7)	**<0.005 ^+µ^**
Bleb needling	0 (0.0)	4 (7.7)	
Major revision	1 (2.8)	15 (28.9)	
Bleb repair	1 (2.8)	3 (5.8)	
**Wound-healing modulators n (%)**			
None	5 (13.79)	20 (38.5)	0.04 ^+$^
MMC	11 (30.6)	12 (23.1)	
5-FU	0 (0.0)	3 (5.8)	
Ologen^®^	0 (0.0)	2 (3.9)	
MMC + Ologen^®^	19 (52.8)	15 (28.9)	
MMC + 5-FU	1 (2.8)	0 (0.0)	

Glaucoma surgeries’ characteristics among patients being prone or not to develop fibrosis, according to clinical appearance. 5-FU = 5-fluorouracil; MMC = mitomycin C; NPDS = non-penetrating deep sclerectomy; ^#^ = χ^2^; ^+^ = Fisher’s exact test; ^$^ = not statistically significant after pairwise comparison using Bonferroni’s correction; ^µ^ = only the “original vs. major revision” comparison among non-fibrosis/fibrosis patients remained statistically significant (*p* = 0.001) after pairwise comparison using Bonferroni’s correction. Numbers in bold show statistically significant values.

**Table 3 ijms-24-11949-t003:** Expression changes of conjunctival genes in fibrosis patients versus non-fibrosis patients.

Gene	∆CT	∆CT	*p*-Value	Fold Change	∆CT	*p*-Value	Fold Change
	Control	Fibrotic			Hyperfibrotic		
*VEGFA*	14.30 ± 0.26	13.66 ± 0.22	0.0617	1.56	11.61 ± 0.21	**0.0001**	**6.47**
*CXCL8*	12.94 ± 0.30	12.22 ± 0.30	0.1009	1.65	10.07 ± 0.48	**0.0003**	**7.32**
*IL18*	8.64 ± 0.16	8.82 ± 0.20	0.5093	-	8.22 ± 0.43	0.3045	-
*TGFA*	7.91 ± 0.40	7.37 ± 0.38	0.3525	-	8.26 ± 1.20	0.7367	-
*TGFB1*	10.71 ± 0.19	10.60 ± 0.21	0.7119	-	10.27 ± 0.39	0.3643	-
*MYC*	14.34 ± 0.28	14.24 ± 0.25	0.7878	-	11.81 ± 0.30	**0.0006**	**5.79**
*THBS1*	10.46 ± 0.27	10.26 ± 0.36	0.6794	-	9.89 ± 0.64	0.4106	-
*CDKN1A*	9.67 ± 0.37	8.58 ± 0.25	**0.0148**	**2.13**	7.03 ± 0.59	**0.0059**	**6.20**
*CDKN2A*	13.78 ± 0.39	13.84 ± 0.21	0.8908	-	14.76 ± 0.50	0.3013	−1.97

Gene expression levels were analyzed in the conjunctiva of glaucoma patients with no signs of fibrosis (control), patients with fibrosis (fibrosis), and patients with early signs of fibrosis (hyperfibrosis). Changes are shown as the mean ± SEM of ΔCTs, followed by *p*-values of each comparison (fibrosis patients versus control patients, hyperfibrosis patients versus control patients, Student’s test). Fold changes were calculated using the 2^−ΔΔCt^ method (see the Section 4). Only fold changes higher than 1.5 or −1.5 are shown. Statistically significant *p*-values are indicated in bold.

**Table 4 ijms-24-11949-t004:** Levels of protein secreted by cultured fibroblasts from fibrosis patients versus non-fibrosis patients.

Protein	pg/mL	pg/mL	*p*-Value
	Control	Hyperfibrotic	
VEGFA	168.81 ± 18.24	596.48 ± 76.06	**0.0007**
IL8	116.05 ± 13.87	792.57 ± 75.30	**˂0.0001**
TGFB1	218.10 ± 16.55	243.03 ± 15.38	0.2995
TSP1	116.37 ± 11.78	109.54 ± 11.47	0.6894

Levels of the VEGFA, IL8, TGFB1, and TSP1 proteins secreted by cultured fibroblasts from control patients and hyperfibrosis patients were measured by ELISA. Changes in protein levels are shown as the mean ± SEM of protein. *p*-values of the comparison between hyperfibrosis patients versus control patients are shown, Student’s test. Statistically significant *p*-values are indicated in bold.

**Table 5 ijms-24-11949-t005:** Analyzed genes and PCR primer sequences.

Gene Symbol	Gene Name	PCR Primer Sequence	
*VEGFA*	Vascular endothelial growth factor	Forward	5′-TGGGTGCATTGGAGCCTTGCCTTGC-3′
		Reverse	5′-GGCAGTAGCTGCGCTGATAGACATCC-3′
*CXCL8*	Interleukin 8	Forward	5′-GCAGAGGGTTGTGGAGAAGT-3′
		Reverse	5′-AACCCTACAACAGACCCACA-3′
*IL18*	Interleukin 18	Forward	5′-CATTGACCAAGGAAATCGGC-3′
		Reverse	5′-CACAGAGATAGTTACAGCCATACC-3′
*TGFA*	Transforming growth factor alfa	Forward	5′-TAATGACTGCCCAGATTCCC-3′
		Reverse	5′-GATGATGAGGACAGCCAGGG-3′
*TGFB1*	Transforming growth factor beta	Forward	5′-CCCAGCATCTGCAAAGCTC-3′
		Reverse	5′-GTCAATGTACAGCTGCCGCA-3′
*MYC*	MYC proto-oncogene	Forward	5′-AAGACTCCAGCGCCTTCTCTC-3′
		Reverse	5′-GTTTTCCAACTCCGGGATCTG-3′
*THBS1*	Thrombospondin 1	Forward	5′-CCTCAGGAACAAAGGCTGCTC-3′
		Reverse	5′-GCCAATGTAGTTAGTGCGGATG-3′
*CDKN1A*	Cyclin-dependent kinase inhibitor 1A	Forward	5′-CCTCATCCCGTGTTCTCCTTT-3′
		Reverse	5′-GTACCACCCAGCGGACAAGT-3′
*CDKN2A*	Cyclin-dependent kinase inhibitor 2A	Forward	5′-CAACGCACCGAATAGTTACGG-3′
		Reverse	5′-AACTTCGTCCTCCAGAGTCGC-3′
*18S*	18S Ribosomal RNA	Forward	5′-GGCGCCCCCTCGATGCTCTTAG-3′
		Reverse	5′-GCTCGGGCCTGCTTTGAACACTCT-3′

## Data Availability

Data available on request from the corresponding author due to privacy restrictions.

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
