# Peer review of "Differential Gene and Protein Expression of Conjunctival Bleb Hyperfibrosis in Early Failure of Glaucoma Surgery"

_ijms, 2023, doi:10.3390/ijms241511949_

Round 1

Reviewer 1 Report (Previous Reviewer 2)

Dear authors,

the manuscript has greatly with your updates and I think it's now suitable for publication

Kind regars

English is good. Only some typos to be checked

Reviewer 2 Report (Previous Reviewer 1)

The authors have revised their manuscript accordingly and provided sufficient explanations for their study design. 

This manuscript is a resubmission of an earlier submission. The following is a list of the peer review reports and author responses from that submission.

Round 1

Reviewer 1 Report

Millá et al. have performed a study on 86 patients to identify signature genes and proteins that are associated with early conjunctival fibrosis and failure of the filtering bleb after glaucoma filtration surgery. To this aim, they compared expression profiles of 9 genes and 4 proteins in conjunctival specimens and supernatants of conjunctival fibroblasts obtained from glaucoma patients with postoperative conjunctival fibrosis (n=41), hyperfibrosis (n=8) and non-fibrosis (n=34).

General comments:

Regarding study design, one issue that should be explicitely stated is the time of sample collection. If tissue specimens were collected at the time of initial glaucoma surgery without knowledge of the outcome, the study design is prospective, as stated. If, however, specimens were collected at the time of surgical revision (page 17: „During each intervention, a block of conjunctival……“), this is a retrospective case-control study that examines suspected risk factors for different outcomes. Even if tissue harvesting was performed at the basal surgery, the majority of fibrosis patients but none of the non-fibrosis patients had previous glaucoma surgery, which introduces a bias and which could imply that the expression alterations in the fibrosis group are secondary due to previous surgical interventions. I suppose that the authors should have performed analyses of covariance to control for confounding effects of previous glaucoma surgery, type of glaucoma surgery, etc. on the outcome of conjunctival fibrosis.

In general, the term „biomarker“ should be avoided in view of the small study numbers (hyperfibrosis n=8), the invasive nature of tissue harvesting, and the lack of assessment of sensitivity and specificity.

Specific points:

1.         Title: The title should be modified, e.g. into „Differential gene and protein expression…“, since the term „profile“ implies an information about all genes/proteins involved in a certain condition.

2.         Methods: The sample numbers for analysis of protein levels are far too low to allow any meaningful conclusions. Moreover, the control samples comprise only 2 non-fibrotic glaucoma and 3 cataract samples, which is not a valid control group for the hyperfibrotic glaucoma study group.

3.         Results: The selection of candidate genes should be explained.

Is there any genotype-phenotype correlation (page 4, line 37)? Since different samples have been analysed for gene and protein expression, the values cannot be correlated.

Figure 1. The means ± standard errors should be illustrated as bar graphs or box plots. In the legend, the n=8 (hyperfibrosis) has to be corrected to n=6.

Gene names should be given according to published guidelines (e.g. CXCL8, etc.)

Reviewer 2 Report

Dear authors,

it was a pleasure for me to review this interesting article. I think that your research is well written and provides useful insights  in the post-operative pathophysiology of the bleb and could open new possibilities in post-operative management. Methods are well presented as well as results. 

The main shortfall of the study is the small number of patients in the hyperfibrosis group, which limits the comparison with other subgroups. Moreover, the introduction section may be slightly summarized, since it's too long as of now.

Compliments for your work, well done.

English is fine. Minor corrections are needed.